# NeuroSafeDrive: An Intelligent System Using fNIRS for Driver Distraction Recognition

**DOI:** 10.3390/s25102965

**Published:** 2025-05-08

**Authors:** Ghazal Bargshady, Hakki Gokalp Ustun, Yasaman Baradaran, Houshyar Asadi, Ravinesh C Deo, Jeroen Van Boxtel, Raul Fernandez Rojas

**Affiliations:** 1Faculty of Science and Technology, University of Canberra, Canberra, ACT 2617, Australia; hakki.ustun@canberra.edu.au (H.G.U.); yasaman.baradaran@canberra.edu.au (Y.B.); raul.fernandezrojas@canberra.edu.au (R.F.R.); 2Institute for Intelligent Systems Research and Innovations (IISRI), Deakin University, Waurn Ponds, VIC 3216, Australia; houshyar.asadi@deakin.edu.au; 3Artificial Intelligence Applications Laboratory, School of Mathematics, Physics and Computing, University of Southern Queensland, Springfield, QLD 4300, Australia; ravinesh.deo@unisq.edu.au; 4Faculty of Health, University of Canberra, Canberra, ACT 2617, Australia; jeroen.vanboxtel@canberra.edu.au

**Keywords:** intelligent transportation systems, fNIRS, driver distraction detection, driving simulator, haemodynamic response

## Abstract

Driver distraction remains a critical factor in road accidents, necessitating intelligent systems for real-time detection. This study introduces a novel fNIRS-based method to to classify varying levels of driver distraction across diverse simulated scenarios, including cognitive, visual–manual, and auditory sources of inattention. Unlike previous work, we evaluated multiple neurophysiological metrics—including oxygenated, deoxygenated, and combined haemoglobin—to identify the most reliable biomarker for distraction detection. Neurophysiological data were collected, and three multi-class classifiers (SVM, KNN, decision tree) were applied across different fNIRS metrics. Our results show that oxygenated haemoglobin outperforms other signals in distinguishing distracted from non-distracted states, while the combined signal performs best in differentiating distraction from baseline. The proposed SVM model achieved ≈ 77.9% accuracy in detecting distracted and relaxed driving states based on brain oxygen levels. Our findings also show that increased distraction correlates with elevated activity in the dorsolateral prefrontal cortex and premotor cortex, whereas driving without distraction exhibits lower neurovascular engagement. This study contributes to affective computing and intelligent transportation systems and could support the development of future driver distraction monitoring systems for safer and more adaptive vehicle control.

## 1. Introduction

Driver distraction is a major cause of traffic accidents. The 2020 Global Status Report indicated that approximately 1.35 million people die from road traffic accidents worldwide each year, with up to 50 million people injured. A significant proportion of road traffic accidents is attributed to distracted driving [1,2]. Distracted driving is defined as any activity that shifts a driver’s focus away from operating a vehicle safely. Driver distraction can be categorised into three types: visual distraction, manual distraction, and cognitive distraction. Among these, cognitive distraction is the most difficult type to address, because it occurs within the driver’s mind [3]. It is difficult to control and is known to reduce attention and cognitive function. Mental workload and cognitive distraction can arise from activities such as using mobile phones, conversing with passengers, and interacting with in-vehicle information systems (e.g., media players and navigation devices). These activities divert attention from the primary task of safe driving, potentially affecting situational awareness and reaction times.

One possible approach to preventing drivers from being distracted is to monitor the driver’s attentional state and predict distraction before it occurs. There are several approaches for identifying driver distraction, including monitoring vehicle kinematics information, using physiological signals such as photoplethysmogram (PPG), electrodermal activity (EDA), electroencephalography (EEG), functional near-infrared spectroscopy (fNIRS), and visual sensors (e.g., camera, eye tracker) to assess driver distraction. Misra et al. (2023) applied PPG, EDA, and eye movement with machine learning classifiers to detect driver distraction [4]. Lei et al. (2025) designed a driver state intelligent monitoring approach using the RES-SE-CNN model for driver distraction detection by processing images captured by dashboard cameras [1]. Several studies have also used EEG to investigate and recognise driver distraction [5].

Functional near-infrared spectroscopy has emerged as a promising tool to monitor driver distraction; fNIRS measures changes in blood oxygenation levels in the brain, which indirectly reflect cognitive workload and mental effort. This makes it particularly useful for detecting increased brain activity associated with secondary tasks or distractions while driving. In particular, it has also been shown that the decrease in attention and cognitive function during driving is reflected in brain activity [6]. It has been observed that fNIRS is particularly effective at detecting prefrontal cortex activity, which is crucial for assessing attention and cognitive workload [7]. EEG is highly sensitive to environmental and muscular artefacts, while fNIRS is less affected by these interferences [8]. In addition, compared to EEG, fNIRS provides higher spatial resolution [9]. Key advantages are offered by fNIRS, including portability and flexibility, making it suitable for real-world driving studies [9]. Given its advantages, fNIRS can be integrated into advanced driver-assistance systems (ADAS) technologies to monitor brain activity in real time. By detecting distractions when they occur, these systems can trigger alerts or required safety actions to refocus attention on the road. This provides a non-invasive method to monitor driver mental workload while driving, enhancing road safety.

Machine learning algorithms provide significant advantages over traditional statistical approaches by enabling real-time, adaptive data analysis. Although several studies have explored the use of fNIRS to detect driver distraction, the application of machine learning in this domain remains relatively limited. Previous research has highlighted the need for further exploration of fNIRS-based distraction detection using machine learning techniques to develop more effective and robust computational models [10,11,12]. Le et al. (2018) investigated cognitive distraction by considering the mental workload of a secondary task. They used a four-channel near-infrared spectroscopy (NIRS) device combined with machine learning methods to estimate cognitive load and predict distraction [12]. Baker et al. (2021), in contrast, used a non-machine learning approach, applying a generalised linear model to analyse driver distraction through fNIRS and eye movement data [13].

This study presents a new approach that aims to classify multiple levels of driver distraction by applying various fNIRS metric analyses with three different machine learning techniques. Specifically, the work utilises fNIRS measurements of oxygenated haemoglobin (ΔHBO2) and deoxygenated haemoglobin (ΔHHB) in order to provide an objective assessment of the driver distraction level. To achieve this objective, we examined various metrics, such as ΔHBO2, ΔHHB, and their combined variations defined as ΔHBO2ΔHHB, to evaluate the potential driver distraction levels that can act as a precursor to harmful traffic incidents. Therefore, we collected high-quality fNIRS data from 21 participants as they drove in a simulated environment under three specific conditions, as follows:Baseline condition: This is a specific period before a stimulus or task begins and is used as a reference point to measure changes in brain activity.Driving-without-distraction condition: This is the period when the driver does not encounter any disturbances.Driving with secondary task-induced distraction: This is when a secondary factor potentially influencing the driver’s focus is introduced.

To implement the above test conditions, each participant completed the driving sessions in three different road scenarios while participating in various secondary tasks, as required, that were also designed to induce distraction. Next, a data-reprocessing procedure was implemented, including a feature-extraction and feature-selection stage to identify the most relevant features for the classification model. We then applied multiple machine learning algorithms to detect driver distraction through binary classification in three comparison parts: (1) baseline vs. driving without distraction (BS vs. SD), (2) baseline vs. driving with distraction (BS vs. DD), and (3) driving without distraction vs. driving with distraction (SD vs. DD).

In accordance with the above test conditions, this research answers the following key hypotheses (HPs):HP1: How does prefrontal cortex activity, particularly in the dorsolateral prefrontal cortex (DLPFC), change under different driving conditions (baseline, driving without distraction, and driving with distraction)?HP2: What haemodynamic response (ΔHBO2, ΔHHB, or a combination) provides the most reliable metric to distinguish between different distractions?

The key contributions of this research are as follows:Introducing a set of optimal features derived from ΔHBO2, ΔHHB, and their combined variations as ΔHBO2ΔHHB across three distinct conditions: baseline, driving without distractions, and driving with a secondary task.Demonstrating that combined variations of oxygenated and deoxygenated haemoglobin are more effective in detecting distracted drivers versus baseline condition, while oxygenated haemoglobin alone is more effective in distinguishing distraction during driving versus driving without distraction.Revealing that bilateral prefrontal activation, particularly in the dorsolateral prefrontal cortex and premotor areas, increases with brain activity during distraction.

This paper is organised as follows: in Section 2, the proposed methods and the dataset are introduced and discussed in detail; the results are explained in Section 3; and, finally, the discussions, future works, limitations, and concluding remarks are provided in Section 4 and Section 5, respectively.

## 2. Materials and Methods

In this section, the details of the simulated distracted driving scenarios under various conditions and measurement techniques are explained. Figure 1 presents a block diagram of the proposed fNIRS-based driver distraction recognition computational model framework. It includes data collection, data preprocessing, feature extraction, feature selection, and machine learning classifiers to perform driver distraction recognition. Further elaboration on the materials and methodology is provided in the following subsections.

### 2.1. Data Collection

In this part, all the important information on data collection is provided, including participants, equipment, and experimental setup. The research, involving human participants, received ethical approval from the Human Research Ethics Committee of the University of Canberra (reference number 14133). The data collection procedure took place at the Human–Computer Interface Laboratory at the University of Canberra, Australia. The experiment for each participant lasted approximately 60 min, including participant training, sensor setup, test execution, and completion of the questionnaire.

#### 2.1.1. Participants

This study involved 21 healthy individuals (i.e., 6 women and 15 men) aged 21 to 62 years with a mean age of 33.9 years. All participants had a full Australian driving license, had no motion sickness or vertigo, and had an average of 11.42 years of driving experience. Individuals known to be prone to vertigo or motion sickness were excluded from the study, due to the potential risk of simulator sickness. Participants received detailed explanations and provided their written informed consent before the start of the experiments. Participants were rewarded $30 for their participation in the study.

#### 2.1.2. Equipment

Figure 2 illustrates the details of the complete setup used in this research project. The following equipment was used for this study:Driving simulator: In this study, a driving simulation system was established. Carnetsoft (Groningen, Netherlands, Research driving simulator software version) was chosen as the driving simulator software for behavioural research [14]. Carnetsoft is suitable for research purposes and has been applied in several studies [4,15,16,17,18]. The simulator included three 27” LCD rendering monitors (one for the left channel, one for the forward view, and one for the right channel) with a resolution of 5760 × 1080, a driver seat, Logitech ( Lausanne, Switzerland) G29 pedals, and steering console, as shown in Figure 2. In this investigation, a customised version of Carnetsoft was applied to Australian road rules and left-hand traffic. A high-performance PC was used to install Carnetsoft and run simulated scenarios with the GeForce RTX 4070 (Santa Clara, CA, USA) specification. Two speakers were used to simulate road noise, wind, tyre sound, engine noise, and distraction tasks. A mobile phone, such as an iPhone X, was used for distraction tasks.A “Brite” head cap (Artinis Medical Systems, Gelderland, the Netherlands) fNIRS sensor with a 24-channel emitter/receiver was used to collect oxygenated haemoglobin and deoxygenated haemoglobin levels, capturing the brain activity of the participants during the driving simulation. During head-cap fitting, we recorded the fNIRS signal to check the signal quality and the light saturation of the optodes. The raw data were collected wirelessly through a Bluetooth module in the OxySoft software (Gelderland, the Netherlands, version 3.5.15.2) from the manufacturer.

#### 2.1.3. Experimental Setups

The developed scenarios aimed to accurately represent real-world distraction conditions and assess the ability of drivers to identify and react to latent hazards. The details of the developed scenarios and experimental setups were as follows. Each participant underwent a 15 min training session on how to use and drive the simulator. Following training, the fNIRS cap was placed on their head, with hair adjustments made to ensure optimal signal quality. The signal was then checked to confirm that the fNIRS system was sending and receiving data correctly.

For all the subjects, the experiment was carried out in three consecutive sets, each with a different type of road (ToR) and distraction techniques, which we call secondary tasks (STs) in this paper. Each set consisted of four phases:Baseline 1 (BS1): A 60 s session in which the participants were asked to sit and relax.Driving without distraction (SD): A 120 s session of driving without distraction. In the driving-without-distraction condition, the participants were instructed to drive normally while focusing solely on navigation and traffic conditions, without added tasks or distractions. They were told to remain attentive to the road environment, follow speed limits, obey traffic signs, and respond to hazards as they appeared in the scenario. During this phase, no secondary tasks or audio instructions were presented, ensuring non-distracted baseline driving performance for comparison.Baseline 2 (BS2): A 60 s session in which the participants were asked to sit and relax.Driving with distraction (DD): A 120 s session of driving with secondary tasks. In the driving-with-distraction condition, the participants were required to perform one of the predefined secondary tasks while driving. These tasks were designed to simulate real-world distractions and are explained in more detail below.

A growing number of technologies, such as cell phones and music players, can significantly distract drivers. Engaging in activities distracts drivers and increases the likelihood of accidents [19]. Devices with highly demanding visual–manual interfaces, such as MP3 players and those imposing high cognitive loads, can impair driving performance [19,20]. The cognitive demand associated with complex music-selection tasks, such as searching for playlists, can further compound the visual demand to read and select songs, resulting in prolonged glances away from the road and reduced driving performance [19]. To examine brain activity during such distractions, we incorporated song selection from a music player as one of the secondary tasks in our driving simulator to assess driver distraction.

To simulate driver distraction scenarios, we designed various distraction approaches based on previous literature reviews. Researchers have found that texting and reading can significantly decrease driving performance and create a significantly higher risk of crashing than when not distracted [21,22]. Moreover, according to the Virginia Tech Transportation Institute, a driver takes his eyes off the road for an average of 4.6 s while reading or sending a text message. Another research carried out in a car simulator revealed that the participants’ reaction time was reduced by approximately 38% while accessing a social network on their smartphones. Research suggests that this kind of distraction is more dangerous than other driving behaviours [21]. A secondary distraction task was designed to distract participants while driving by requiring them to read text messages on a mobile phone attached to a holder beside the driver.

Researchers have shown that the number of driving errors is highest in passenger conversations and that the complexity of dialogue for both individuals decreases as traffic demands increase. In addition, driving conditions have a direct influence on the complexity of the conversation, thus mitigating the potential negative effects of a conversation on driving [23]. In this study, a true/false-based conversation was designed to simulate distracted driving by replicating a two-way conversation between the driver and a virtual interlocutor.

Therefore, in the distracted driving scenarios, the participants performed one of the three secondary tasks (STs) while driving. We used three different types of distraction tasks, including true/false questions, selecting a specific song from a music player, and reading a series of short message service (SMS) texts. These tasks were randomly assigned on different types of roads to introduce distractions while driving:ST 1: Answering true/false questions was developed as a cognitive distraction task in a spoken task. A series of statements were played through the speakers after a constant period of five seconds, to which the driver responded. This secondary task acted as an alternative to the conversations with passengers or hands-free mobile phone calls that were made while driving vehicles.ST 2: The participants were instructed to select a specific song from the music list available in the Spotify music app on a mobile phone. Every 20 s, a series of unsorted alphabetic letters was played through the speakers. The driver was then required to remove one hand from the steering wheel and select a song from the list that started with the designated letter. Each letter corresponded to only one song in the list. This secondary task served as an alternative to using an in-car media player while driving, allowing evaluation of cognitive, manual, and visual distractions.ST 3: A series of notifications of short text messages were received on the mobile phone every eight seconds while driving, each accompanied by a notification sound. The participants were expected to read the messages silently. To ensure that they had read the texts, they were asked to recall key details about the message content after the experiment. This secondary task was designed to assess cognitive and visual distraction.

At the end of each driving scenario, the participants were asked to complete a questionnaire that included the situation awareness rating technique (SART) [24] and the Susceptibility to Driving Distraction Questionnaire (SDDQ) [25] to measure performance during scenarios. Examples are as follows: The situation was challenging to me; I was focused on driving only; I was concentrating on the secondary task only; I was concentrating on the two tasks (both the driving and secondary tasks); I obtained all the information from the secondary tasks; I was fully concentrating on various traffic patterns, signs, and other conditions; I was familiar with the situation; I felt more anxious than usual when driving at night; I made an effort to look for potential hazards; I was disturbed by thoughts of an accident or car crash. The questionnaires used a 5-point Likert scale, ranging from 1 (strongly disagree) to 5 (strongly agree).

The start and end of each phase were marked by beep sounds. The participants completed all three sets according to a randomisation order. Depending on the randomisation order, the non-distraction scenario for a given road type sometimes preceded the corresponding distraction scenario, while in other cases the order was reversed. The order of the three sets varied, in terms of road types and distraction techniques, which were also randomised.

The scenarios were designed on various types of roads, from urban to highway, to explore different driving patterns. Each type of road featured a predefined speed limit, it was conveyed through in-simulation signage, and it was communicated to the participants before each drive. There was a navigation aide for driving, and the drivers had to follow the direction of the road navigator. In each scenario, a latent hazard was embedded within a specific road section, defining a critical zone where the detection and response of the hazard were evaluated. These scenarios spanned three distinct types of road: (a) ToR 1: urban, (b) ToR 2: urban with intersections, and (c) ToR 3: highway at night. They were randomly chosen to capture diverse driving patterns. We followed related articles in the literature on the development of road types [4,26,27]. Figure 3 shows some examples of the road hazards applied for each ToR, and it is followed by descriptions of road types:
ToR 1: During this two-minute urban driving scenario, the driver navigates various challenges while maintaining a speed limit of 50 km/h. They start in a city setting with pedestrians and crosswalks, which requires paying attention to pedestrian movements. Further along, two parked cars occupy the left lane, demanding awareness of potential hazards. The scenario also includes pedestrians and animals as obstacles, requiring the driver to stop when necessary. Additional challenges involve traffic lights, parked cars along the road, and varying speed limits ranging from 40 to 60 km/h, which require constant situational awareness and adherence to traffic regulations.ToR 2: During this urban driving scenario, the driver navigates intersections, heavy traffic, and roundabouts while adapting to changing road conditions. The presence of multiple vehicles requires careful positioning in the lane and awareness of the surrounding traffic. As the route continues, the driver approaches a four-way stop-controlled intersection where visibility is significantly restricted, due to large trucks and stop signs obscured by vegetation. This requires extra caution, requiring the driver to slow down, check for other vehicles, and proceed only when it is safe. Each challenge appears sequentially, ensuring continuous engagement with the dynamic urban environment.ToR 3: During this night-time highway driving scenario, the driver maintains a speed of 80 km/h while adhering to Australian road rules. The route includes merging highways and exit points, requiring careful lane changes and awareness of surrounding traffic. Other vehicles on the highway exhibit different driving behaviours, including slow driving, speeding, and overtaking, creating a dynamic environment. The driver has to navigate traffic safely, adjusting their speed and position while ensuring that they overtake only when permitted.

In the following, we used the following setup for the fNIRS sensor, as changes in ΔHBO2 and ΔHHB concentrations (μ mol/L) were recorded using a continuous-wave wireless fNIRS device. The system consisted of 24 channels placed on the prefrontal cortex. It utilised 10 sources and 8 detectors, with optodes spaced 35 mm apart on the frontal lobe. Near-infrared light, emitted at 760 nm and 840 nm, was sampled at a frequency of 50 Hz.

### 2.2. Data Preprocessing and Feature Extraction

After completing the data collection, we applied data cleaning, preprocessing, and feature engineering. The data were segmented and labelled into three classes: baseline, driving without distraction, and driving with distraction. As described in the experiment setup subsection, for each subject the sequence followed this order: BS1 was recorded first, followed by driving without distraction; then BS2 was recorded; and, finally, driving with distraction was applied. Each subject repeated this three times for different ToRs.

In the data analysis, only BS1 observations were used in each ToR per subject for the baseline class, and BS2 was excluded from this analysis. The three observations for driving without distraction and the three observations for driving with distraction per subject were applied in the data analysis. Consequently, the dataset consisted of a total of (3 baseline (BS1) × 60 s × 50 Hz) + (3 driving without distraction × 120 s × 50 Hz) + (3 driving without distraction × 120 min × 50 Hz) = 45,000 samples.

To suppress noise and physiological pulsations in the fNIRS data, each available fNIRS channel was processed using a 4th-order Butterworth low-pass infinite impulse response filter with a cut-off frequency of 0.16 Hz [28]. During fNIRS data acquisition, there can be various common noise sources that affect the measurements. Therefore, they were removed by applying the filter.

We extracted features from various window sizes as non-overlapping 1, 10, 20, 30, and 60 s based on three fNIRS sub-datasets: 24 channels of ΔHBO2, 24 channels of ΔHHB, and 48 combined channels of ΔHBO2HHB. Mean value features of ΔHBO2 and ΔHHB are commonly used in fNIRS studies [29,30]. The mean value of fNIRS signals (ΔHBO2, ΔHHB, and ΔHBO2HHB) provides a robust measure of brain activity in a specific time window. As shown in Equation (Equation 1), the mean is calculated by averaging all sample values within a defined window:(1)$$\mu =\frac{1}{N}\sum_{i=1}^{N}x_{i}$$

Mean value is easy to interpret compared to higher-order statistical features, such as variance or skewness. In addition, since fNIRS signals are often noisy, using only the mean helps avoid overfitting and ensures a simplified and effective representation of brain activity [31,32]. Therefore, in this paper, we applied the mean values for all channels in three fNIRS metrics of ΔHBO2, ΔHHB, and ΔHBO2HHB across various window sizes (1, 10, 20, 30, and 60 s).

### 2.3. Feature Selection

Feature selection (FS) is crucial for improving model efficiency by focusing on important features, reducing dimensionality, and, ultimately, enhancing overall performance in machine learning tasks. Feature selection is a common preprocessing step in machine learning that selects an informative subset of features to improve classification performance [33]. Joint mutual information (JMI) is based on the principles of information theory, entropy, and mutual information. Joint entropy is usually used to measure information content shared by multiple variables. Assuming that *X* and *Y* are two random variables, (X,Y) denotes the corresponding joint random variable, p(x,y) denotes its probability distribution, and the joint entropy H(X,Y) measures the information of the variables *X* and *Y*. The number of variables is generalised to multiples [34]. The joint entropy of a joint variable composed of multiple random variables $X_{1},X_{2},\ldots ,X_{n}$ can be expressed as follows in Equation (Equation 2):(2)$$H\left(X_{1},\ldots ,X_{n}\right)=-\int_{x_{1}}\cdots \int_{x_{n}}p\left(x_{1},\ldots ,x_{n}\right)\log p\left(x_{1},\ldots ,x_{n}\right)\;dx_{n}\ldots dx_{1}.$$

In this paper, the features were extracted from fNIRS signals collected across 24 channels of ΔHBO2, 24 channels of ΔHHB, and 48 channels of ΔHBO2HHB. Each channel corresponded to a spatial location on the prefrontal cortex or other targeted brain regions. The features represented mean values calculated within a fixed window size of 30 s with no overlap to segment the fNIRS signals. The selected features through the JMI-based approach (measured by joint mutual information), reflected both temporal summaries (mean over a duration) and spatial coverage (across different brain regions/channels). We tested multiple window sizes in this study and determined that a window size of 30 s was the optimal segmentation.

### 2.4. Machine Learning Classifiers

Changes in brain activity associated with cognitive tasks, such as driving or a driver performing a secondary task that can lead to distraction, can be monitored by fNIRS. Studies have shown that an increase in cognitive load, as measured by fNIRS, may correlate with poorer driving performance and increased driving infractions [35]. In this paper, we investigated how fNIRS can be used to predict the degree of driving distraction based on brain activity patterns. Driver distraction can be classified into three classes: baseline (BS), which represents the resting state or minimal cognitive load, providing a reference for brain activity without task engagement; driving without distraction (SD), which captures brain activity during normal driving conditions, which can indicate the cognitive load required to drive alone; and driving with distraction (DD), which measures the additional cognitive load imposed by a secondary task, highlighting changes in brain activation due to distraction.

To achieve this, all the features from the ΔHBO2, ΔHHB, and ΔHBO2HHB signals were applied to train and test well-known classifiers such as support vector machine (SVM), K-nearest neighbour (KNN), and decision trees (DTs) to identify distraction levels using the fNIRS data. We employed Bayesian parameter optimisation, carefully tuning the classifiers to achieve the best results. The models were developed, trained, and tested using different window lengths: 1, 10, 20, 30, and 60 s. After that, JMI was used to select a reduced feature set consisting of mean value features extracted from the ΔHBO2, ΔHHB, and ΔHBO2HHB signals. Then, the machine learning classifiers mentioned above were applied to the selected features, using the optimal window size (30 s) to increase the accuracy of the classifiers.

The evaluation was carried out in the three subsets corresponding to ΔHbO2, ΔHHb, and ΔHBO2HHB to predict the degree of distracted driving based on brain activity patterns. To assess the performance of these machine learning models, 7-fold cross-validation was applied on the binary classification tasks: BS vs. DD, BS vs. SD, SD vs. DD. The 7-fold cross-validation was evaluated using a leave-subject-out cross-validation approach by leaving out three subjects as the test set and using the remaining subjects as the training dataset. This process was repeated seven times, ensuring that each subject served as the test set exactly once. Table 1, Table 2 and Table 3 show the best hyperparameters applied to each machine learning classifier in BS vs. SD, BS vs. DD, and SD vs. DD.

Performance metrics consisting of accuracy, F1 score, sensitivity, and specificity were averaged to provide a comprehensive assessment of the overall model performance. Furthermore, we systematically tested the identification of the best-performing model with varying numbers of features based on their JMI rank.

## 3. Results

The analysis of window size effects on model performance showed that a window size of 30 s consistently led to better outcomes across the different metrics tested.

### 3.1. Reference Values (Before Feature Selection)

Table 4, Table 5 and Table 6 present the performance of different classifiers (SVM, KNN, DT) across various metrics for the three binary classifications: BS vs. DD, BS vs. SD, and SD vs. DD, based on the extracted features for all channels. The best results are highlighted in bold. As shown in the following tables, the results comparing feature extraction across different classifiers demonstrated that ΔHBO2HHB outperformed ΔHBO2 and ΔHHB in BS vs. DD and BS vs. SD, with accuracy of 75.20% and 72.24%, respectively, in the SVM classifier. However, only ΔHBO2 outperformed ΔHBO2HHB and ΔHHB in SD vs. DD, with accuracy of 58.75 % in the SVM classifier.

### 3.2. Feature Selection Results

In this subsection, the results after using feature selection are presented. Based on JMI rankings, the proposed machine learning algorithms were trained and tested on selected features for all three metrics (ΔHBO2HHB, ΔHBO2, ΔHHB) in BS vs. DD, BS vs. SD, and ΔHBO2 for SD vs. DD. The ranked features were sequentially incorporated into the models, starting from the highest-ranked and progressively adding more (e.g., the top two, top three, and so on). This approach was used to evaluate the performance of each feature subset and to identify the optimal combination of features. Table 7 indicates the effective features selected based on the JMI algorithm rankings for each classification condition.

Figure 4 presents the performance of the machine learning models (SVM, KNN, DT) for each set of ranked feature subset. As shown, the highest accuracy (77.9%) for BS vs. DD was achieved using the top 24 ranked features of ΔHBO2HHB in SVM. For BS vs. SD, SVM performed best with the top 32 features, while DT achieved the same highest accuracy (73.95%) with the top 25 features. In contrast, for SD vs. DD, the highest accuracy (61.77%) was obtained using the top 14 ranked features of ΔHBO2 in the SVM classifier.

The results show that applying the JMI feature-selection technique improved performance across all metrics. The achieved results demonstrate that SVM outperforms both KNN and DT.

Table 8, Table 9 and Table 10 indicate the results obtained for three classifiers (SVM, KNN, and DT) after applying the JMI feature-selection algorithm for ΔHBO2HHB, ΔHBO2, and ΔHHHB in BS vs. DD, BS vs. SD, and SD vs. DD.

### 3.3. Cortical Activation

Figure 5 illustrates the activation map showing variation in brain workload for distracted drivers. The analysis is presented for three binary conditions using the best-performing haemoglobin concentration set: (a) baseline versus driving with distraction, (b) baseline versus driving without distraction, (c) driving without distraction versus driving with distraction.

In the ΔHBO2 map, the blue regions indicate lower ΔHbO2 levels, suggesting reduced oxygenated blood flow. The green-to-yellow regions represent moderate increases in ΔHbO2, likely reflecting active cognitive involvement. The red regions indicate the highest ΔHbO2 levels, potentially corresponding to increased neural activity during distraction, reflecting cognitive load or compensatory process.

In the ΔHHb map, the blue regions indicate lower ΔHHb levels, which generally correlate with increased oxygenation and neuronal activity. The green-to-yellow regions show moderate ΔHHb levels, indicating varying oxygen consumption. The red regions indicate high ΔHHb concentration, suggesting reduced oxygen delivery or local deactivation.

Figure 6 shows the spatial mapping of the fNIRS channels placed in the prefrontal cortex, including ventrolateral prefrontal cortex (VLPFC), DLPFC, medial prefrontal cortex (mPFC) and orbitofrontal cortex (OFC), to represent the coverage of the regions of interest.

## 4. Discussion

In this research paper, an fNIRS-based computational machine learning model was developed and evaluated to recognise driver distraction. Three different distraction techniques and three different types of roads were simulated in a virtual driving simulator. The results indicate that the combination of oxygenated and deoxygenated haemoglobin provides a more robust interpretation of brain activity in distraction scenarios, particularly when comparing baseline conditions to driving with and without distraction. However, recognising the difference between driving without distraction and driving with distraction remains a challenge.

Figure 6 shows the spatial profiles of the fNIRS channels with probes placed in the prefrontal area. The channels included are represented according to their location in specific brain regions: VLPFC, DLPFC, medial prefrontal cortex, and orbitofrontal cortex. The neural mechanisms involved in adapting to driving distractions are illustrated in Figure 5, section (a). The neural mechanisms of adaptation to driving distractions involved increased ΔHbO2 concentration in the mPFC, particularly the DLPFC and premotor areas, indicating greater mental effort for cognitive control and attention regulation. The ΔHHB map shows significant variations, with bilateral reduction suggesting increased cognitive participation and right-lateralised activation highlighting asymmetric attentional processing. These findings reinforce the role of neurovascular coupling in maintaining cognitive performance during distraction.

Section (b) indicates that driving without distraction compared to baseline, although it indicates an increase in ΔHbO2 that shows the brain actively maintains situational awareness and motor planning while driving, shows lower-intensity activation compared to distracted driving, indicating a reduced cognitive load. The ΔHHB map shows efficient oxygen use with a balanced workload between hemispheres, aligning with the road safety findings that distractions increase cognitive demand.

Section (c) illustrates how machine learning analysis identified ΔHBO2 as the primary indicator of cognitive load, showing increased activation in the dorsolateral prefrontal cortex and premotor areas during distraction. Overall, channels 11 and 16 were located in the medial prefrontal cortex region, as illustrated in Figure 6. This area is known to play a critical role in sustained attention, self-referential processing, decision making, and cognitive control, all of which are core cognitive demands during driving, especially when distraction or mental workload is involved. The consistently higher ΔHBO2 values observed in channels 11 and 16 in all three driving conditions, BS vs. SD, BS vs. DD, and SD vs. DD, suggest that these channels capture increased neurovascular activity associated with increased attentional regulation and cognitive effort in the mPFC. This finding aligns with previous neuro-imaging studies, which reported that the mPFC, particularly its dorsal regions, activates significantly during tasks requiring executive control and monitoring, such as multitasking, managing distractions, and maintaining focus on the road in cognitively demanding driving scenarios.

This study contributes to affective computing and intelligent transportation systems by providing an objective method to detect driver distraction. The JMI-selected features and regions with high ΔHbO2 activation in Figure 6 demonstrate that the JMI method effectively identifies neurally meaningful and task-relevant channels. This further supports the interpretability and utility of our feature-selection approach for driver distraction detection. The JMI applied in this paper improved the feature-selection strategy and enhanced classification accuracy, particularly in distinguishing distraction from baseline conditions. The integration of fNIRS and machine learning offers a promising approach to future adaptive driver assistance systems, enhancing road safety by enabling proactive monitoring of distractions.

In terms of comparing our results with previous studies, there are only a limited number of investigations focused on driver distraction detection using fNIRS and machine learning in simulated environments. Therefore, a direct comparison of our findings with previous research is not feasible at this stage. The studies cited in the Introduction section differ significantly, in terms of methodology, making it challenging to align their results with ours. For example, Le et al. (2018) employed fNIRS to detect driver attention levels under naturalistic driving conditions, classifying mental workload into three categories (low, medium, and high). Their model, evaluated across different fNIRS channels, achieved classification accuracy of 82.90% and prediction accuracy of 82.74% [12]. The other studies referenced in the Introduction section did not incorporate both machine learning and fNIRS signals, and, therefore, a meaningful comparison with our work is not possible.

This study has several limitations that should be acknowledged. First, for safety and ethical reasons, it was not possible to apply the proposed model in real-time on-road driving scenarios. Instead, data were collected in a simulated environment, which may not have fully replicated the complexity and unpredictability of real-world driving. Additionally, while we tested the model in three driving scenarios, the task diversity could be further expanded to include more realistic and varied sources of distraction. In this study, only data from 21 subjects were used, which could be considered a limitation, and in the future we will extend the experiment with more participants.

In future work, we plan to increase the sample size and include data from a broader demographic. We will also incorporate multimodal data from various physiological and behavioural sources to enhance model robustness. Furthermore, we plan to explore multi-feature extraction approaches to improving performance and robustness via different features. Advanced deep learning and machine learning algorithms will also be explored, with a focus on improving generalisation through subject-independent evaluation methods such as leave-one-subject-out cross-validation. Moreover, future studies should examine the model’s applicability across different road types and simulated driving environments to increase ecological validity.

Ultimately, the findings of this study lay the groundwork for real-time driver monitoring systems that could contribute to improved road safety and better situational awareness for both drivers and traffic management systems.

## 5. Conclusions

This study developed a distracted driving simulation that incorporated various types of roads and distraction techniques to detect driver distraction using fNIRS signals. We evaluated three machine learning classifiers for multiclass classification and applied multiple fNIRS signal metrics (ΔHBO2, ΔHHB, and ΔHBO2HHB), along with a JMI-based feature-selection technique to enhance performance.

Our findings show that combining ΔHBO2 and ΔHHB (ΔHBO2HHB) improved classification accuracy in most class comparisons, particularly in differentiating the baseline from distracted states. However, ΔHBO2 alone slightly outperformed the combined metric in distinguishing between driving without distraction and distracted driving. Among the classifiers, support vector machine achieved the highest accuracy (77.9%) in detecting distracted versus relaxed driving states.

The results also indicate that driving with distraction induces a higher cognitive load, as evidenced by increased ΔHBO2 in the dorsolateral prefrontal cortex and the premotor cortex. Moreover, ΔHBO2 proved to be a more sensitive biomarker than ΔHHB for identifying changes in attentional and cognitive effort.

These insights contribute to the growing field of real-time driver monitoring and suggest practical applications in affective computing and advanced driver assistance systems to improve road safety.

## Figures and Tables

**Figure 1 sensors-25-02965-f001:**
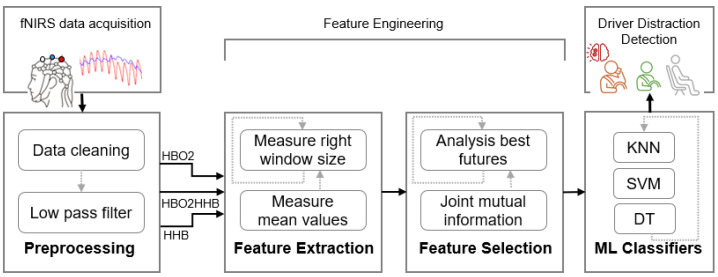
Framework of the fNIRS-based driver distraction recognition computational modelling.

**Figure 2 sensors-25-02965-f002:**
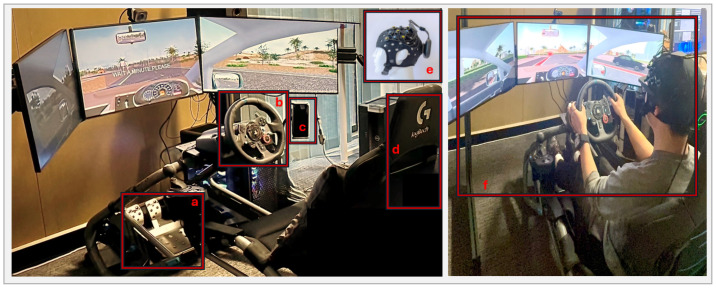
A driving simulator was established to detect driver distraction. The elements of the simulator are identified as (**a**) pedals (**b**) steering wheel (**c**) mobile phone (**d**) seating chair (**e**) fNIRS head cap; (**f**) visualises a sample experiment applied by the established simulator in the lab.

**Figure 3 sensors-25-02965-f003:**
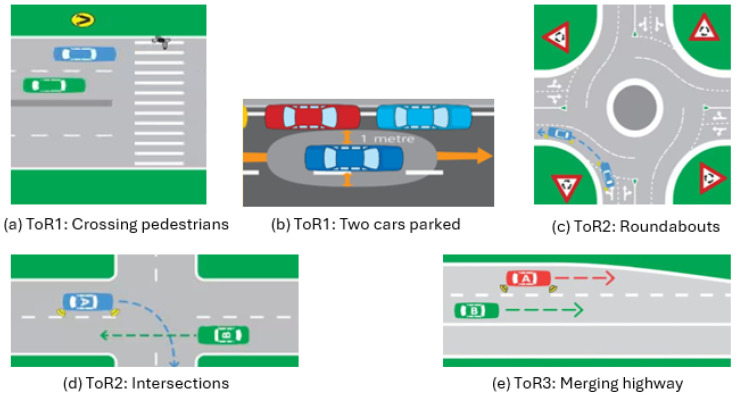
Visualisations of some of the road hazards described above, applied to different scenarios.

**Figure 4 sensors-25-02965-f004:**
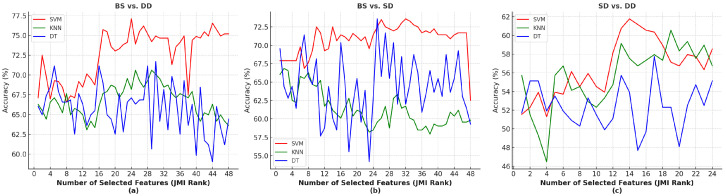
Performance of selected features based on JMI algorithm for three machine learning algorithms and three various binary classifications in (**a**) BS vs. DD in ΔHBO2HHB (**b**) BS vs. SD in ΔHBO2HHB (**c**) SD vs. DD in ΔHBO2.

**Figure 5 sensors-25-02965-f005:**
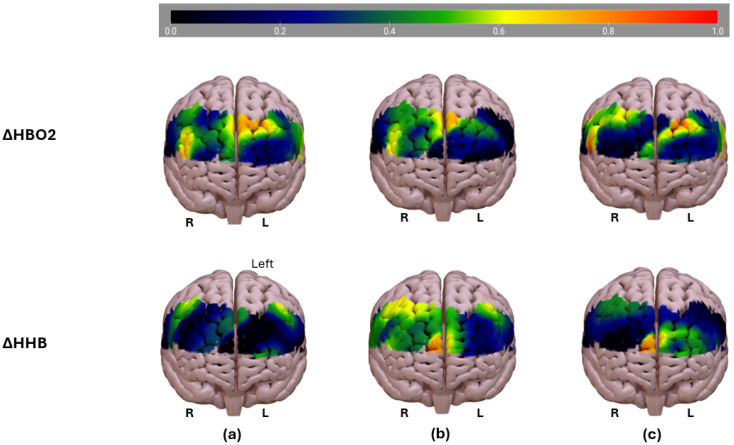
Activation maps of oxygenated haemoglobin (ΔHbO2, top row) and deoxygenated haemoglobin (ΔHHb, bottom row) across three condition comparisons: (**a**) baseline vs. distracted driving, (**b**) baseline vs. driving without distraction, (**c**) baseline vs. distracted driving. Positive values (warmer colours) indicate higher activation. The left and right hemispheres of the brain are labelled as “L” and “R”, respectively, for orientation. For visualisation purposes only, the original activation data (ranging from −1 to 1) were linearly rescaled to a 0–1 range, using min–max normalisation. This transformation preserved the relative differences and distribution of the activation values, ensuring a clearer and more interpretable graphical representation in Surfice.

**Figure 6 sensors-25-02965-f006:**
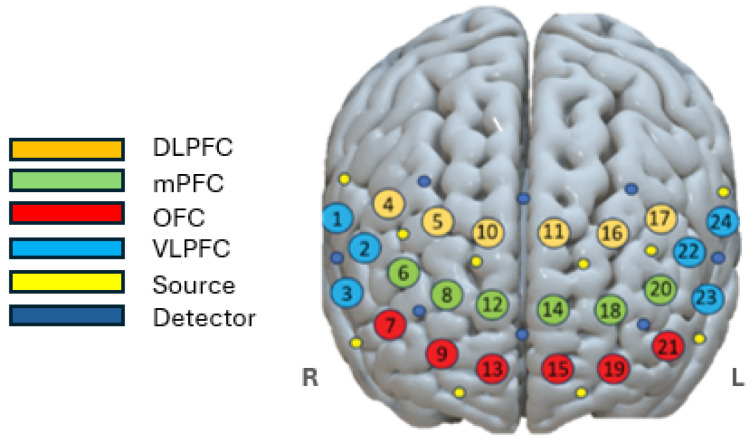
The spatial profiles of the fNIRS channels were mapped, with probes placed on the prefrontal area including VLPFC, DLPFC, mPFC, and OFC. The left and right hemispheres of the brain are labeled as “L” and “R”, respectively, for orientation.

**Table 1 sensors-25-02965-t001:** Hyperparameters for three machine learning classifiers, SVM, KNN, and DT, for three modalities in BS vs. SD.

Models	Metrics	Configurations	Values
SVM	ΔHBO2, ΔHHB, ΔHBO2HHB	C	1
		G	0.1
		kernel	rbf
		iteration	10,000
KNN	ΔHBO2	number of neighbours	20
		weights	uniform
	ΔHHB	number of neighbours	25
		weights	uniform
	ΔHBO2HHB	number of neighbours	27
		weights	uniform
DT	ΔHBO2	criterion	entropy
		max_depth	5
		min_samples_split	2
		min_samples_leaf	12
		max_features	sqrt
		random_state	42
	ΔHHB	criterion	entropy
		max_depth	5
		min_samples_split	2
		min_samples_leaf	10
		max_features	sqrt
		random_state	42
	ΔHBO2HHB	criterion	entropy
		max_depth	5
		min_samples_split	2
		min_samples_leaf	18
		max_features	sqrt
		random_state	42

**Table 2 sensors-25-02965-t002:** Hyperparameters for three machine learning classifiers, SVM, KNN, and DT, for three modalities in BS vs. DD.

Models	Metrics	Configurations	Values
SVM	ΔHBO2, ΔHHB, ΔHBO2HHB	C	1
		G	0.05
		kernel	rbf
		iteration	10,000
KNN	ΔHBO2, ΔHHB, ΔHBO2HHB	number of neighbours	25
		weights	uniform
DT	ΔHBO2, ΔHHB, ΔHBO2HHB	criterion	entropy
		max_depth	5
		min_samples_split	2
		min_samples_leaf	20
		max_features	sqrt
		random_state	42

**Table 3 sensors-25-02965-t003:** Hyperparameters for three machine learning classifiers, SVM, KNN, and DT, for three modalities in SD vs. DD.

Models	Metrics	Configurations	Values
SVM	ΔHBO2	C	1
		G	0.05
		kernel	rbf
		iteration	10,000
	ΔHHB, ΔHBO2HHB	C	1
		G	0.1
		kernel	rbf
		iteration	10,000
KNN	ΔHBO2, ΔHHB, ΔHBO2HHB	number of neighbours	15
		weights	uniform
DT	ΔHBO2	criterion	entropy
		max_depth	10
		min_samples_split	5
		min_samples_leaf	14
		max_features	sqrt
		random_state	42
	ΔHHB	criterion	entropy
		max_depth	10
		min_samples_split	5
		min_samples_leaf	10
		max_features	sqrt
		random_state	42
	ΔHBO2HHB	criterion	entropy
		max_depth	10
		min_samples_split	5
		min_samples_leaf	2
		max_features	sqrt
		random_state	42

**Table 4 sensors-25-02965-t004:** Performance of different classifiers (SVM, KNN, DT) across various metrics for the three binary classifications: BS vs. DD.

Metrics	Classifiers	Accuracy	F1-Score	Sensitivity	Specificity
		(in %)	(in %)	(in %)	(in %)
ΔHBO2	SVM	70.35	78.48	82.54	44.54
	KNN	68.46	68.26	75.40	53.78
	DT	67.66	66.84	77.38	47.06
ΔHHB	SVM	63.34	70.90	72.62	43.70
	KNN	58.49	57.93	54.36	67.23
	DT	59.03	58.59	66.67	42.86
Δ**HBO2HHB**	**SVM**	**75.20**	**81.55**	**83.73**	**57.14**
	KNN	63.61	68.01	57.94	75.63
	DT	64.42	63.10	72.22	47.90

Bold font shows the best performance.

**Table 5 sensors-25-02965-t005:** Performance of different classifiers (SVM, KNN, DT) across various metrics for the three binary classifications: BS vs. SD.

Metrics	Classifiers	Accuracy	F1-Score	Sensitivity	Specificity
		(in %)	(in %)	(in %)	(in %)
ΔHBO2	SVM	62.53	72.14	75.00	36.13
	KNN	59.84	59.66	65.08	48.74
	DT	59.30	59.70	60.32	57.14
ΔHHB	SVM	67.93	76.40	81.75	38.65
	KNN	59.57	58.91	63.49	51.26
	DT	60.38	60.56	66.67	47.06
Δ**HBO2HHB**	**SVM**	**72.24**	**80.81**	**89.28**	**36.13**
	KNN	60.11	65.16	60.32	59.66
	DT	63.88	64.26	65.48	60.51

Bold font shows the best performance.

**Table 6 sensors-25-02965-t006:** Performance of different classifiers (SVM, KNN, DT) across various metrics for the three binary classifications: SD vs. DD. performance.

Metrics	Classifiers	Accuracy	F1-Score	Sensitivity	Specificity
		(in %)	(in %)	(in %)	(in %)
Δ**HBO2**	**SVM**	**58.75**	**56.47**	**55.10**	**62.30**
	KNN	56.74	56.08	46.94	66.27
	DT	55.13	54.25	48.16	61.90
ΔHHB	SVM	51.91	49.91	51.02	52.78
	KNN	52.72	50.11	39.19	65.87
	DT	54.73	54.25	46.94	62.30
ΔHBO2HHB	SVM	54.93	56.52	61.22	48.81
	KNN	55.74	42.12	33.88	76.98
	DT	58.55	57.83	63.67	53.57

Bold font shows the best performance.

**Table 7 sensors-25-02965-t007:** List of the effective features selected based on JMI ranking for optimised metrics for each class BS vs. DD, BS vs. SD, and SD vs. DD in ascending order from higher rank to lower rank.

Classification	Effective Selected Features
BS vs. DD	ΔHHB 16, ΔHHB 7, ΔHHB 1, ΔHBO2 24, ΔHHB 12, ΔHHB 4, ΔHHB 10, ΔHBO2 23, ΔHHB 19, ΔHBO2 19, ΔHBO2 8, ΔHBO2 17, ΔHBO2 12, ΔHHB 13, ΔHBO2 13, ΔHBO2 22, ΔHBO2 18, ΔHBO2 4, ΔHBO2 6, ΔHBO2 15, ΔHHB 21, ΔHBO2 7, ΔHHB 2, ΔHHB 5
BS vs. SD	ΔHHB 8, ΔHHB 17, ΔHBO2 19, ΔHHB 1, ΔHHB 7, ΔHHB 5, ΔHHB 10, ΔHBO2 5, ΔHBO2 12, ΔHHB 14, ΔHBO2 9, ΔHHB 4, ΔHHB 24, ΔHBO2 20, ΔHHB 12, ΔHHB 19, ΔHBO2 14, ΔHHB 6, ΔHHB 18, ΔHBO2 7, ΔHBO2 6, ΔHBO2 23, ΔHHB 16, ΔHHB 12, ΔHBO2 11, ΔHHB 15, ΔHBO2 16, ΔHHB 13, ΔHBO2 18, ΔHBO2 8, ΔHHB 2, ΔHHB 22
SD vs. DD	ΔHBO2 15, ΔHBO2 7, ΔHBO2 19, ΔHBO2 9, ΔHBO2 23, ΔHBO2 13, ΔHBO2 24, ΔHBO2 1, ΔHBO2 10, ΔHBO2 14, ΔHBO2 2, ΔHBO2 18, ΔHBO2 4, ΔHBO2 21

**Table 8 sensors-25-02965-t008:** Performance of different classifiers (SVM, KNN, DT) across various metrics for the three binary classifications: BS vs. DD after feature selection.

Metrics	Classifiers	Accuracy	F1-Score	Sensitivity	Specificity
		(in %)	(in %)	(in %)	(in %)
ΔHBO2	SVM	72.13	80.81	83.46	52.37
	KNN	70.61	69.21	78.02	57.31
	DT	69.12	68.75	79.42	52.48
ΔHHB	SVM	65.26	72.33	74.14	46.77
	KNN	61.32	59.23	58.66	69.54
	DT	62.45	61.24.59	68.76	45.89
**ΔHBO2HHB **	**SVM**	**77.09**	**83.07**	**86.11**	**57.98 **
	KNN	70.62	70.79	71.43	68.91
	DT	71.70	69.63	85.72	42.02

Bold font shows the best performance.

**Table 9 sensors-25-02965-t009:** Performance of different classifiers (SVM, KNN, DT) across various metrics for the three binary classifications: BS vs. SD after feature selection.

Metrics	Classifiers	Accuracy	F1-Score	Sensitivity	Specificity
		(in %)	(in %)	(in %)	(in %)
ΔHBO2	SVM	65.32	75.42	79.31	40.79
	KNN	63.67	62.53	67.55	52.2
	DT	63.9	62.14	63.42	60.04
ΔHHB	SVM	69.21	78.98	85.40	42.54
	KNN	61.7	59.11	64.46	52.62
	DT	61.44	62.49	68.7	49.22
**ΔHBO2HHB**	**SVM**	**73.59**	**81.11**	**85.32**	**48.74**
	KNN	66.85	61.95	88.89	20.17
	**DT**	**73.59**	**71.08**	**85.71**	**47.90**

Bold font shows the best performance.

**Table 10 sensors-25-02965-t010:** Performance of different classifiers (SVM, KNN, DT) across various metrics for the three binary classifications: SD vs. DD after feature selection.

Metrics	Classifiers	Accuracy	F1-Score	Sensitivity	Specificity
		(in %)	(in %)	(in %)	(in %)
**ΔHBO2**	**SVM**	**61.77**	**59.43**	**58.78**	**64.68**
	KNN	60.56	60.05	51.43	69.44
	DT	59.96	58.40	60.41	59.52
ΔHHB	SVM	53.23	52.67	53.87	54.72
	KNN	54.62	53.47	42.25	68.12
	DT	58.73	56.5	47.34	61.67
ΔHBO2HHB	SVM	56.32	58.23	63.45	50.21
	KNN	57.44	45.26	39.81	78.86
	DT	60.35	59.43	66.11	55.86

Bold font shows the best performance.

## Data Availability

The data presented in this study are available on request from the corresponding author, due to privacy.

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
