# Peer review of "NeuroSafeDrive: An Intelligent System Using fNIRS for Driver Distraction Recognition"

_sensors, 2025, doi:10.3390/s25102965_

Round 1
Reviewer 1 Report
Comments and Suggestions for Authors
- There are many statistical features, such as mean, variance, slope, integral value, and gravity value. Why are only the mean chosen? How about the performance of the other features?
- Why choose the 7-fold cross-validation instead of the leave-one-subject-out cross-validation method?
- What temporal-spatial characteristics do the selected features based on JMI and what is the optimal window size?
- Why is there only one result graph in Figure 5(c)? Please explain the reason.
- What does Figure 6 illustrate, and it was not mentioned in the paper?
- The language expression in this manuscript should be further condensed and improved by professional support, such as ‘Lei el at. (2025) were designed an intelligent driver state’ from Line 47 to Line 48.
- Discuss the limitations and future research directions of the study.
Reviewer 2 Report
Comments and Suggestions for Authors
In this paper, a new approach is presented that aims to classify multi-levels of driver distraction by applying various fNIRS metrics analysis with three different machine learning techniques. This study contributes towards affective computing and intelligent transportation systems. This paper is interesting, while there are some problems should be considered as follows.
(1) In abstract is too long, and the novelty of this paper is not clear in the section.
(2) In the section of Experimental Setups, more details about how to make Driving without distraction and Driving with distractions need be added.
(3) In the section of Data preprocessing and Feature extraction, some relevant formula could be added.
(4) In the section of Discussion, the authors should state and provide an association of the results with the literature and the results reported by previous research.
(5)The conclusion is tedious, thus it should be simplified.
Reviewer 3 Report
Comments and Suggestions for Authors
Please see the attached file.

Round 2
Reviewer 1 Report
Comments and Suggestions for Authors
Many thanks to the Authors for the revised version.
All the major issues were sufficiently addressed.
Author Response
Comment 1: All the major issues were sufficiently addressed.
Response 1: Thank you so much for your valuable comments.
Reviewer 3 Report
Comments and Suggestions for Authors
Figure 5: Could the authors double-check if the scale bar in Figure 5 is from 0-1, which does not start from a negative value? Meaning that in all three comparisons, all channels have positive values compared to the baseline condition?
Author Response
Comment 1: Figure 5: Could the authors double-check if the scale bar in Figure 5 is from 0-1, which does not start from a negative value? Meaning that in all three comparisons, all channels have positive values compared to the baseline condition?
Response 1:
Thank you for your comment regarding the scaling of activation values in the heatmaps. We would like to clarify that this adjustment was necessary due to the visualization limitations of the Surf Ice software (version 1.0.20211006+; www.mricro.com). In this version, values of 0 and below are rendered using the same colour on the surface map, making it impossible to distinguish between areas of low negative activation and regions with no activation.
To overcome this constraint and enable a continuous, interpretable colour gradient for both negative and positive activation values, we linearly rescaled the original activation data from a range of -1 to 1 to a range of 0 to 1 prior to visualization only for this purpose. This rescaling preserved the relative differences and distribution of the data, ensuring a more accurate graphical representation of brain activation patterns. Importantly, this adjustment was applied solely for visualization purposes, and all statistical analyses were performed on the original values.
Addressing: We have now clarified this point in the manuscript to prevent any confusion. Please see the highlighted section in the Figure 5 caption.